# Comprehensive analysis of CMTM family and immune infiltration in esophageal carcinoma

Liying Xue[1]*, Shuting Gou[1], Yu Zhang[1], Ruirui Yuan[1], Chang Dong[2], Rongyao Hao[1], Na An[1], Xianghong Zhang[1], Jie Li[2]*

1 Laboratory of Pathology, Hebei Medical University, Shijiazhuang, China, 2 Department of Hematology, Hebei General Hospital, Shijiazhuang, China

* doclijie@163.com (JL), xueliying123@163.com, xueliying@hebmu.edu.cn (LX)

## Abstract

### Objective

Esophageal carcinoma (ESCA) is one of the most common malignant diseases and contributes to the annual burden of death worldwide. A better understanding of the underlying molecular changes is urgently required to identify early diagnostic biomarkers and effective therapeutics. The chemokine-like factor (CKLF)-like MARVEL transmembrane domain-containing family (CMTMs) is reported to be entangled in many human cancers. However, the role of CMTMs in ESCA remains unclear.

### Methods

The differential expressions of CMTMs between ESCA and normal tissues were analyzed using TCGA database. The relationships between CMTMs and immune infiltration in the tumor microenvironment (TME) were also evaluated to explore their underlying values in the diagnosis and prognosis of ESCA.

### Results

The results showed that ESCA showed significantly higher expressions of CMTM1,3,6,7 and lower expressions of CMTM4,5 than normal tissue ($P < 0.05$). Meanwhile, CMTM3,4,8 expressions were correlated with the tumor stage of ECSA patients. The analysis on immune infiltrations (CD8+ T, Tregs, NK and macrophages) showed that M2 macrophages was dominant in TME, with significantly higher levels than the other cells (F = 326.93, $P < 0.001$). The higher abundance of M2 macrophages and Tregs significantly shortened the survival time of patients with ESCA ($P = 0.01$). Interestingly, the expression levels of CMTM1,3,5,7 were comparable to the abundance of M2 macrophages (CMTM1: r = 0.172168; CMTM3: r = 0.313221; CMTM5: r = 0.130669; CMTM7: r = 0.119922; $P < 0.05$). CMTM2,4,5,7,8 positively correlated with Tregs ($P < 0.05$). Moreover, we found positive associations between the expression of CMTMs and the signatures of M2 macrophages (MS4A4A, VSIG4 and CD163).

**Data availability statement:** All relevant data are within the manuscript and its Supporting information files.

**Funding:** This study was supported by the Natural Science Foundation of Hebei Province (No. H2021206395).

**Competing interests:** The authors have declared that no competing interests exist.

## Conclusion

There were differential expressions of CMTMs between ESCA and normal tissues. Furthermore, the expression of CMTMs was positively correlated with M2 macrophages, indicating a possibility that CMTMs may become a new immunotherapy target for ESCA.

## 1. Introduction

Esophageal carcinoma (ESCA) is one of the most common malignant diseases, with approximately 604 thousand new cases and 544 thousand deaths worldwide annually [1]. Commonly, ESCA has two histologic subtypes: esophageal squamous cell carcinoma (ESCC) and esophageal adenocarcinoma (EAC), which have different etiologies and geographic variations. In some high-risk areas in Asia (especially in China), the incidence of ESCC is declining, possibly due to economic gains and dietary improvements, whereas 90% of ESCA patients have ESCC [1]. Advanced ESCC have a low 5-year survival rate with high invasion and metastases, even after surgery or chemotherapy. Therefore, a better understanding of the underlying molecular changes is urgently required to identify early diagnostic biomarkers and effective therapeutics.

The chemokine-like factor superfamily (CKLFSF) is a recently discovered gene family consisting of CKLF and CKLFSF 1-8 [2]. Due to having a MARVEL (MAL and related proteins for vesical trafficking and membrane link) domain, CKLFSF1-8 is renamed CKLF-like MARVEL transmembrane domain containing 1-8 (CMTM1-8). CMTM1, containing a C-c motif, shows higher sequence identity with chemokines, while CMTM8 has 39.3% amino acid similarity with transmembrane-4 superfamily 11 (TM4SF11). CMTM2-7 interestedly displays biological characteristics between chemokines and TM4SF [3,4]. CMTMs are broadly expressed in human tissues, and are entangled in many kinds of diseases. Currently, due to their structural and functional similarity to classic chemokines and TM4SF, these genes have attracted much interest in cancer research. CMTM family members are regarded as potential prognostic markers for multiple human cancers possibly due to their differential expressions between tumor and normal tissues [5]. Especially, CMTM4 and CMTM6 are considered to be potential therapeutic targets because they enhance PD-L1 expression via reducing PD-L1 ubiquitination in tumor cells [6,7]. However, the role of CMTMs in ESCA and their relationships with immune infiltration in tumor microenvironment (TME) remains unclear.

In this study, we investigated the expressions of CMTMs and the relationship between CMTMs and immune infiltration in ESCA. We aimed to better understand the underlying values of CMTM family members as potential molecular therapeutic targets and biomarkers in ESCA through performing an integrated bioinformatics analysis. We found that the expression levels of some CMTM members were significantly related to immune infiltrating cells in TME. Therefore, CMTMs may become a new immunotherapy target for ESCA.

## 2. Materials and methods

### 2.1. Patients and samples

One hundred and eighty-five samples of ESCA and 286 normal tissues were analyzed using Gene Expression Profiling Interactive Analysis (GEPIA, http://gepia.cancer-pku.cn/), which includes the mRNA data of 9,736 tumors from The Cancer Genome Atlas (TCGA, http://ualcan.path.uab.edu/) [8,9] and 8,587 normal samples from the Genotype-Tissue Expression (GTEx) project. ESCA samples (n = 185) were used to evaluate the relationship between CMTMs and immune infiltration collected in TIMER2.0 (http://timer.cistrome.org/) [10,11].

## 2.2. Differential expression of CMTMs between ESCA and normal tissue

The expression of CMTMs in ESCA was analyzed using TCGA according to bioinformatics analysis [12,13]. Furthermore, we also investigated the relationship between CMTMs expression and tumor stage to predict their impact on patient prognosis.

## 2.3. Immune infiltration characters of ESCA

The immune infiltration characteristics of ESCA were evaluated using GEPIA2021 (http://gepia2021.cancer-pku.cn/). We performed the quantitative comparison of the proportion among cell types, including CD8 + T, regulatory cells (Tregs), natural killer (NK) cells, macrophages M1, and macrophages M2 in ESCA using CIBERSORT. The immunophenotypic signature for TAM and Tregs in many common malignant tumors, especially ESCA, was also analyzed in GEPIA2 based on a given set of TCGA. The transmembrane protein MS4A4A and V-set and immunoglobulin domain containing 4 (VSIG4), expressed specifically on TAMs [14], are regarded as immunophenotypic signature with CD163. The results were shown in the form of heatmaps using GraphPad Prism 9.0 software. Furthermore, we separated patients with ESCA (TCGA) into two groups according to the proportion of M2, Tregs, and both. The Kaplan–Meier curve for each group was plotted with the survival data. The difference between the curves was statistically measured using the log-rank test.

## 2.4. The relationships between CMTMs and immune infiltration in ESCA

The relationship between the expression of CMTMs and immune infiltrating cells in ESCA tissues was evaluated using TIMER2.0 (http://timer.cistrome.org/). TIMER includes several algorithms, including TIMER, xCell, EPIC, CIBERSORT, CIBERSORT-ABS, and QuanTIseq. Immune-infiltrating cells, including macrophages, CD4 + T cells, CD8 + T cells, regulatory cells (Tregs), natural killer (NK) cells, myeloid dendritic cells (MDC), and myeloid-derived suppressor cells (MDSC) were analyzed using the right deconvolution tool. The data were plotted as heatmaps using GraphPad Prism 9.0 software (GraphPad Software, Inc., La Jolla, CA, USA).

## 2.5. Statistical analysis

Statistical analyses for differential expression between ESCA were performed using Graph-Pad Prism 9.0 software. Multiple comparisons were performed using a one-way analysis of variance (ANOVA). GEPIA 2, TIMER 2.0, and TCGA database also provided t-test analysis. Survival curves were statistically analyzed using the log-rank test. To account for multiple comparisons in our large-scale data analyses, the Benjamini-Hochberg procedure was applied to control the False Discovery Rate (FDR). $P < 0.05$ was considered statistically significant.

## 3. Results

### 3.1. Differential expressions of CMTMs in ESCA

CMTM family members were differently expressed in ESCA tumor tissues, compared with normal samples (Fig 1A). ESCA showed significantly higher expressions of CMTM1,3,6,7 and lower expressions of CMTM4,5 than normal tissue ($P < 0.05$, FDR-adjusted). However, CMTM2,8 expressions at mRNA level showed no significant difference between ESCA and normal tissue ($P > 0.05$). Meanwhile, CMTM3,4,8 expression was associated with the tumor stage of ESCA patients (Fig 1B). Further analysis based on TCGA database showed that CMTM3 was higher expressed in stageII, III than in stage I ($P < 0.05$), while CMTM8 was higher expressed in stage III than in stage II ($P < 0.05$, S1 Fig).

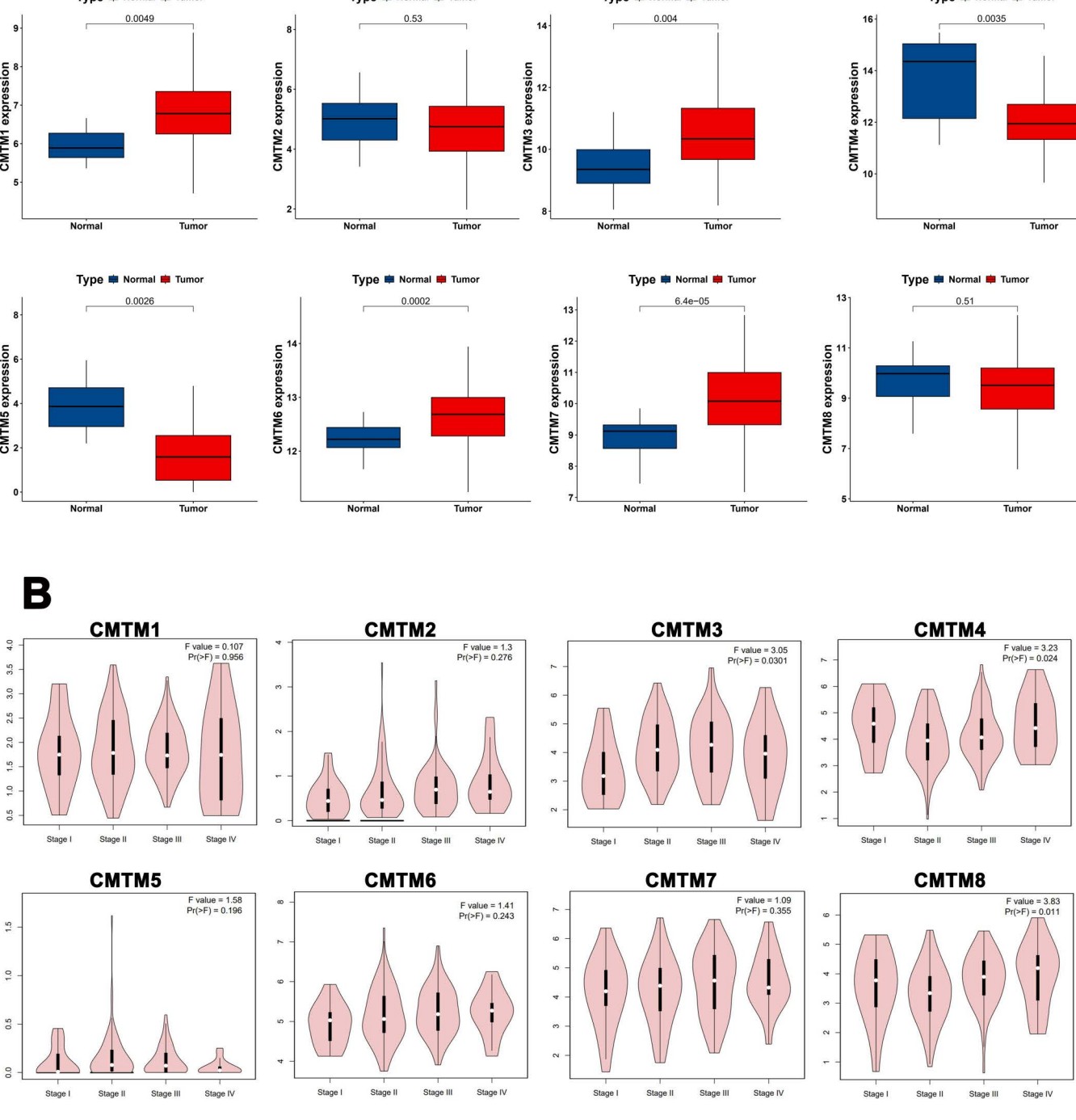

**Fig 1. The expressions of CMTMs in ESCA and normal esophageal tissue based on TCGA.** (A) The expressions of CMTMs in ESCA, compared with the normal tissues (*P*: FDR-adjusted); (B) The relationships between CMTMs expression and tumor stage.

A comparative analysis of CMTMs expression was carried out between the two histologic subtypes based on TCGA database. The results showed that there were statistically significant differences between ESCC and EAC. As shown in Fig 2, CMTM1, 2, 4, 6, 7 and 8 were highly expressed in EAC, while CMTM3 and 5 were highly expressed in ESCC (*P* < 0.05, FDR-adjusted).

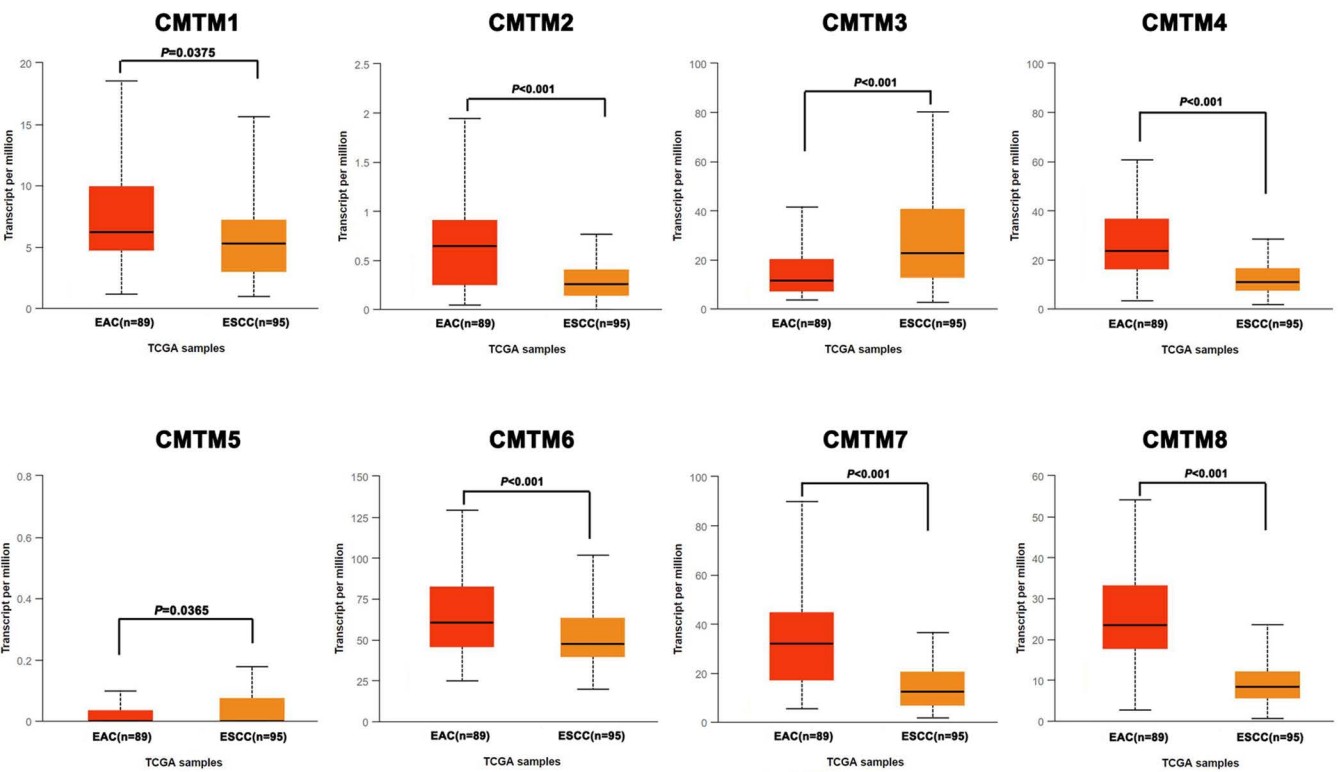

**Fig 2. The differential expressions of CMTMs between ESCC and EAC (*P*: FDR-adjusted).**

## 3.2. Associations of clinical characteristics and CMTMs with mortality risk in ESCA patients

We further investigated the associations between clinical characteristics and CMTMs and mortality risk in ESCA patients using univariable and multivariable analyses. As showed in S1 Table, males had a significantly higher risk of death compared to females, with an odds ratio (OR) of 3.41 in univariable analysis ($P = 0.034$) and 3.83 in multivariable analysis ($P = 0.047$). Compared to Stage I, patients with Stage III and Stage IV disease had higher odds of death with ORs of 4.18 ($P = 0.038$) and 37.33 ($P = 0.003$) in univariable analysis, respectively. These results suggest that gender, tumor stage, and lymph node metastasis are significant predictors of survival in ESCA. Among the CMTM family members, CMTM2 and CMTM8 showed significant associations with survival in univariable analysis. However, these associations were not significant in multivariable analysis, indicating that their effects may be mediated through interactions with other clinical or molecular factors.

## 3.3. The higher proportion of M2 macrophages and Tregs shortens the survival time of ESCA patients

Numerous studies have demonstrated that immune cell infiltration is linked to the occurrence, growth, prognosis, and response to immunotherapy in many human carcinomas. Therefore, we analyzed the immune infiltrations, including CD8 + T cells, regulatory cells (Tregs), natural killer (NK) cells, and macrophages in ESCA. The data showed that in TME, macrophages M2 were dominant, with significantly higher levels than in the others (F = 326.93, $P < 0.001$, Fig 3A). Furthermore, the markers of macrophage subtypes, including

monocytes, TAMs, M1, and M2 macrophages, were also analyzed in several common cancers, such as ESCA, HNSC, STAD, and others. As shown in Fig 3B, the TAM markers were highly detected, especially for the M2 macrophage subtype. Although the proportion of M2 macrophages and Tregs in each class of cells did not affect the overall survival of patients with ESCA, a higher proportion of both types could significantly shorten their survival time ($P=0.01$, Fig 3C).

### 3.4. The association between CMTMs and immune infiltrations in ESCA

The relationship between CMTM family members and immune infiltration was quantified using TIMER. We found that the CMTMs showed different links (positive, negative, or no) with various immune infiltrating cells in the heat map (Fig 4). The expression of CMTMs showed a negative correlation with the abundance of Th1 CD4 + T cells, naive CD8 + T cells, and MDSC but was positively correlated with M2 macrophages, naive CD4 + T cells, and Tregs (except for CMTM3 and Tregs). Interestingly, except for CMTM3, which was negatively correlated with the abundance of Tregs, the other members, CMTM2,4,5,7,8, showed a positive correlation with Tregs ($P<0.05$). Importantly, the expression levels of CMTM1,3,5,7 was comparable to the abundance of M2 macrophages (CMTM1: r = 0.172168; CMTM3:

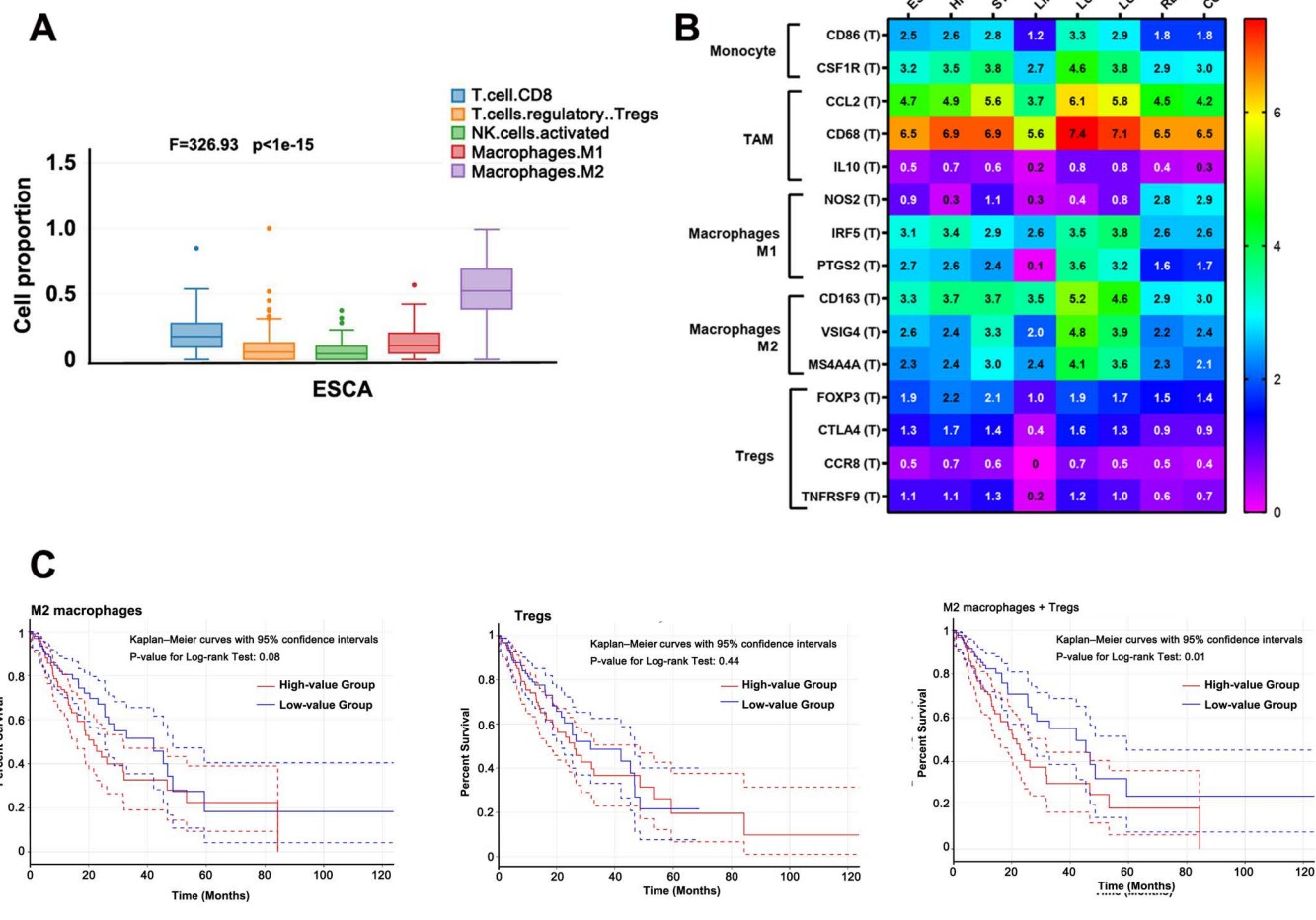

**Fig 3. Immune infiltration characters of ESCA.** (A) The proportion of the principal immune infiltrating cells in TME. (B) The immunophenotypic markers of macrophages and Tregs in common cancers. (C) The survival time and the proportion of M2 macrophages and Tregs in TME.

r = 0.313221; CMTM5: r = 0.130669; CMTM7: r = 0.119922; $P < 0.05$). Further analysis of the signatures of M2 macrophages showed positive associations between the expression of CMTM1,3,4,5,6,7 and MS4A4A, CMTM3,5,6,7 and VSIG4, and CMTM1,3,4,5,6,7 and CD163 ($P < 0.05$, Fig 5). These findings demonstrated that CMTMs must be inextricably linked to immune infiltration, especially M2 macrophages.

## 4. Discussions

Previous studies have demonstrated that CMTMs are involved in cell proliferation, apoptosis, and metastasis and function as tumor suppressors or tumor promoters in multiple human cancers. CMTMs are also potentially regarded as diagnostic markers because of their differential expression between tumor and normal tissues. Using bioinformatics analysis, the data showed that CMTMs are involved in the occurrence of ESCA, indicating the possibility that some CMTM members may be diagnostic markers for ESCA.

### 4.1. CMTM1 may be a potential diagnostic marker in ESCA

CMTM1,2 has a highly tissue-specific distribution and is mostly expressed in testis tissue. Most recently, the differential expression of CMTM1,2 between tumor and normal tissues has attracted more attention. For example, the isoform of CMTM1-v17 is highly expressed in many human tumors such as breast cancer and hepatocellular carcinoma (HCC) [15,16]. CMTM1-v17 prevents TNF-α-induced apoptosis by activating the NF-kB pathway and promoting cellular proliferation in breast cancer [15]. The isoform of CMTM1-v5, interacting with calcium-modulating cyclophilin ligands, negatively regulates the Ca2 + response and then induces caspase activation, resulted in cell apoptosis [17]. CMTM2 expression was reduced and related to epithelial-mesenchymal transition (EMT) in HCC [18]. Choi et al. revealed that higher CMTM2 expression in gastric cancer is closely

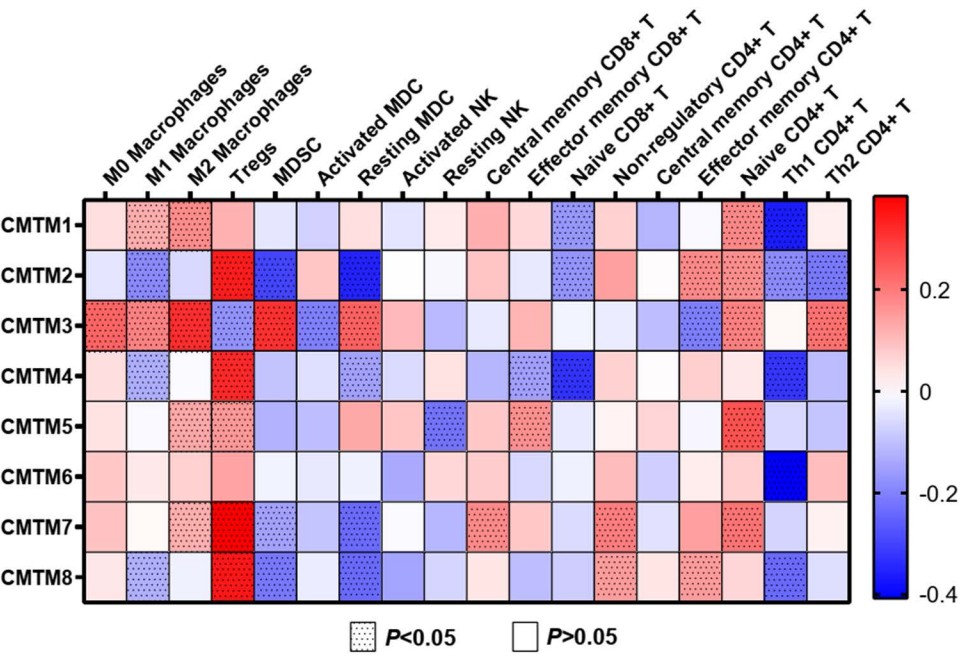

**Fig 4. The relationships between the expressions of CMTMs and immune infiltrating cells in ESCA using TIMER 2.0 (http://timer.cistrome.org/).**

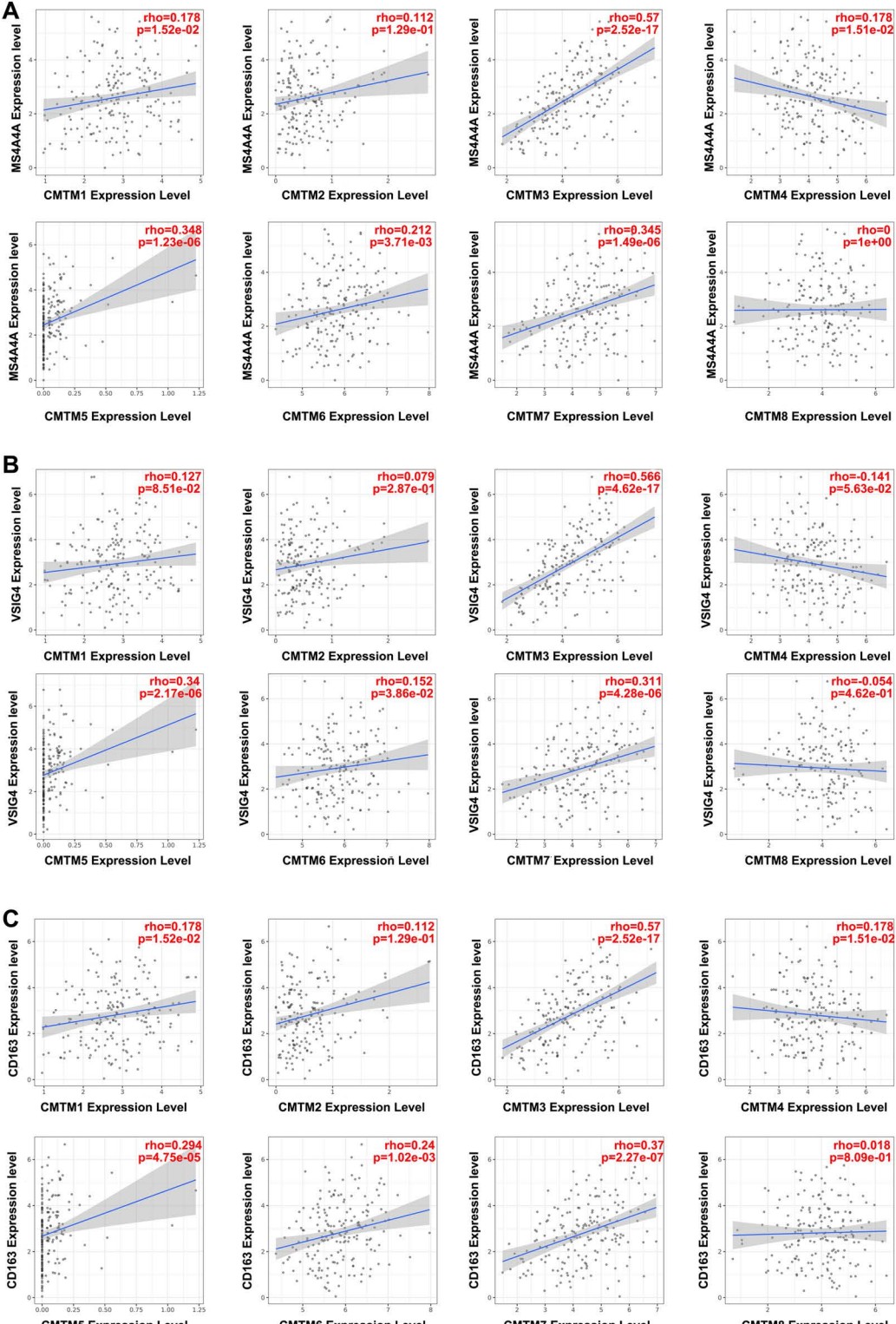

**Fig 5. The expressions of CMTMs were related with the signatures of M2 macrophages using TIMER 2.0 (TCGA).**
CMTMs have relationships with MS4A4A (A), VSIG4 (B), and CD163 (C).

associated with better overall survival [19]. In this study, the expression of CMTM1 was significantly higher in ESCA than in normal tissue, indicating that it may serve as potential diagnostic marker for ESCA.

### 4.2. The roles of CMTM3,5 in ESCA

CMTM3 and CMTM5, which have 42% homology and contain a typical CpG island, generally tend to be regarded as tumor suppressors because of their low expression in gastric cancer, prostate cancer, and HCC [20–23]. For example, CMTM5-v1 inhibits cell growth and migration by downregulating EGFR signaling in prostate cancer [22]. However, in pancreatic cancer, CMTM3 promotes cell proliferation and migration and plays a tumor-promoting role [24]. In this study, CMTM3 was highly expressed in ESCA, and was closely related to the tumor stage, while CMTM5 was lowly expressed in tumor tissue. More research is needed to explore the roles of CMTM3,5, a tumor-suppressor or promoter, in ESCA.

### 4.3. CMTM4,6 may serve as predictive markers of anti-PD-L1 treatment in ESCA

CMTM4 and CMTM6, which share 55% amino acid identity, are downregulated in clear cell renal cell carcinoma and lung cancer and act as tumor suppressors [25–27]. However, some studies have revealed an upregulated expression of CMTM6 in HCC, CRC and head neck squamous cell carcinoma (HNSCC), facilitating the development of malignant diseases [28–31]. For example, CMTM6 overexpression promotes cell proliferation, invasion, and migration and induces EMT by stabilizing vimentin in HCC [28]. In 2017, CMTM6 is co-localized with PD-L1 at the plasma membrane, and can increase PD-L1 protein half-life through reducing PD-L1 ubiquitination in cancer cells [6]. Notably, CMTM6 deletion significantly decreases the inhibition of tumor-specific T cell activity by reducing PD-L1 expression [7]. Meanwhile, CMTM6/PD-L1 has a potential relationship with tumor recurrence, perhaps because of its ability to regulate inflammatory cell density [32,33]. A recent study demonstrated that CMTM6 drives cisplatin resistance by regulating Wnt signaling through the ENO-1/AKT/GSK3β axis [31]. These findings provide insights into the functions of CMTM4 and CMTM6 and highlight a therapeutic target to overcome the immune escape of tumor cells. In this study, the results showed that CMTM6 was highly expressed in ESCA. Given the participation of CMTM4, 6 in PD-L1 stabilization, the level of CMTM6 expression is likely a predictive marker for ESCA patient response to anti-PD-L1 treatment.

### 4.4. The roles of CMTM7,8 in ESCA

Relative to CMTM4,6, studies related to CMTM7,8 in human cancer are slightly fewer. Even so, the downregulated expression of CMTM7,8 has been detected in HCC [34] and NSCLC [35]. CMTM8 downregulation induces EMT-like processes through HGF/c-MET/ERK signaling in HCC cells [34]. In ESCC, CMTM7 was silenced or downregulated in 44.4% of cell lines, including EC109, KYSE410, and KYSE180 [36]. Meanwhile, CMTM7 overexpression inhibited KYSE410 and KYSE180 cell growth by inducing G1/S arrest and suppressing EGFR-PI3K/AKT signaling in KYSE180 cells *in vitro*. In contrast, our analysis showed that CMTM7 was upregulated in ESCA. To find the reason for the opposite result, we further compared the expressions of CMTM family members between ESCC and EAC. The data showed different expressions of CMTMs between both histological subtypes. The expressions of CMTM7 and CMTM8 were significantly lower in ESCC than in EAC. Hence, it is necessary to clarify the roles of CMTM7,8 in ESCA through a large number of experiments.

## 4.5. The relationships between CMTMs and immune infiltration

We further investigated the associations between clinical characteristics and CMTMs and survival time in ESCA patients. We found that gender, tumor stage, and lymph node metastasis are significant predictors of survival in ESCA. Among the CMTM family members, CMTM2 and CMTM8 showed significant associations with survival in univariable analysis. However, these associations were not significant in multivariable analysis, indicating that CMTM family members could contribute to ESCA progression indirectly, potentially by influencing immune regulation, TME, or other signaling pathways. For example, CMTM6 expression positively correlates with the immune infiltrating level of M2 macrophages, T cells, CD4 memory T cells, and CD4 memory resting T cells in TME, resulted in regulating antitumor responses [37,38]. Most recently, Miao et al reported that the presence of CMTM6 and CD58 on tumor cells significantly affects T cell-tumor interactions and response to PD-L1/PD-1 blockade [39]. Therefore, CMTM6 is a promising target for developing immunotherapy. Meanwhile, CMTM4 can facilitate escape from T cell-mediated cytotoxicity by stabilizing PD-L1 expression in HCC [40]. Inhibition of CMTM4 not only suppressed HCC cell growth but also increased CD8 + T-cell infiltration in immunocompetent mice. In addition, the expression of CMTM3 was positively correlated with macrophages, dendritic cells, and CD4 + T cells in grade II and III gliomas [41]. In contrast, the expression of CMTM8 was negatively correlated with CD8 + T cells, dendritic cells, and NK cells in colon cancer [42]. Like CMTM6, CMTM7 was shown to be a new lead candidate for regulating PD-L1 in breast tumors undergoing EMT [43].

It is widely known that immune infiltrations are inextricably associated with the occurrence, growth, prognosis, and response to immunotherapy in human carcinomas. A suggestive inverse correlation was detected between CD8 T cells and CD163 + M2 macrophages, suggesting that the abundance of TAMs might predict the scarcity of tumor-infiltrating T cells [44]. In this study, we found that M2 macrophages were dominant in ESCA, and a higher proportion of both M2 macrophages and Tregs significantly shortened the survival time of patients. CMTMs showed different links (positive, negative, or no) with various immune infiltrating cells. In particular, the expression of CMTMs showed a negative correlation with the abundance of Th1 CD4 + T cells, naive CD8 + T cells, and MDSC but mainly showed a positive correlation with M2 macrophages, naive CD4 + T cells, and Tregs. These findings indicate that the up- or down-regulation of CMTMs must be involved in immune infiltration and then affect the antitumor immune response in ESCA.

This study has several limitations. First, our findings are based on publicly available datasets. While we used rigorous computational methods, these variations could impact the generalizability of our results. Second, our predictions lack experimental validation. Although we employed robust statistical approaches, further laboratory experiments are needed to confirm the biological relevance of our findings. However, these limitations do not diminish the value of our findings as a starting point for further investigation.

In conclusion, CMTM family members show different expression profiles between ESCA and normal tissues, indicating that CMTMs play important roles in the development and progression of ESCA. In particular, the expressions of CMTMs showed a positive correlation with M2 macrophages, naive CD4 + T cells, and Tregs. It is speculated that CMTMs may become a new immunotherapy target. Further research on epigenetic changes, post-translational protein modifications, and related signaling pathways for CMTMs are required in ESCA. Most likely, CMTMs family members could contribute novel ideas and targets for the diagnosis and treatment of ESCA.

## Supporting information

**S1 Fig. The relationships between CMTMs expression and tumor stage.** * $P < 0.05$, compared with normal tissue; # $P < 0.05$, compared with stage I; & $P < 0.05$, compared with stage II. (TIF)

**S1 Table. Associations of clinical characteristics and CMTMs with mortality risk in ESCA patients.**
(DOCX)

## Author contributions

**Conceptualization:** Liying Xue.

**Data curation:** Liying Xue, Jie Li.

**Formal analysis:** Liying Xue, Jie Li.

**Funding acquisition:** Liying Xue.

**Investigation:** Shuting Gou, Yu Zhang, Ruirui Yuan.

**Methodology:** Shuting Gou, Yu Zhang, Ruirui Yuan, Chang Dong, Rongyao Hao, Na An.

**Project administration:** Liying Xue.

**Resources:** Shuting Gou, Yu Zhang, Ruirui Yuan, Na An.

**Software:** Chang Dong, Rongyao Hao.

**Supervision:** Xianghong Zhang.

**Validation:** Liying Xue, Jie Li.

**Visualization:** Rongyao Hao, Na An.

**Writing – original draft:** Liying Xue, Jie Li.

**Writing – review & editing:** Liying Xue, Jie Li.

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
