## [Decision Letter · Decision Letter 0]

16 Dec 2024

PONE-D-24-50157Comprehensive analysis of CMTM family and immune infiltration in esophageal carcinomaPLOS ONE

Dear Dr. Xue,

Thank you for submitting your manuscript to PLOS ONE. After careful consideration, we feel that it has merit but does not fully meet PLOS ONE’s publication criteria as it currently stands. Therefore, we invite you to submit a revised version of the manuscript that addresses the points raised during the review process.

**Please kindly prepare a detail point-by-point response to each comment from the reviewers. Thank you. **

We look forward to receiving your revised manuscript.

Kind regards,

Dave Mangindaan

Academic Editor

PLOS ONE

**Journal Requirements:**

Natural Science Foundation of Hebei Province 

Reviewers' comments:

Reviewer's Responses to Questions

**Comments to the Author**

1. Is the manuscript technically sound, and do the data support the conclusions?

Reviewer #1: Yes

Reviewer #2: Yes

Reviewer #3: Partly

2. Has the statistical analysis been performed appropriately and rigorously? 

Reviewer #1: Yes

Reviewer #2: Yes

Reviewer #3: No

3. Have the authors made all data underlying the findings in their manuscript fully available?

Reviewer #1: Yes

Reviewer #2: Yes

Reviewer #3: No

4. Is the manuscript presented in an intelligible fashion and written in standard English?

Reviewer #1: Yes

Reviewer #2: Yes

Reviewer #3: No

5. Review Comments to the Author

**Reviewer #1:**  Overall, the paper described a very interesting statistical analysis between ESCA and the expressions of CMTM. As far as the methodology goes, I believe most have been conducted really well and the findings are greatly insightful. However, I have a few inquiries as follows.

1. "CMTM3,4,8 expression was associated with the tumor stage of ESCA patients"

While the p-values indicate that the differences shown in Fig. 1B is significant, the differences in the violin plot does not seem to be too noticeable for certain classes. For example, CMTM3 and CMTM4 on tumor stages 2 and 3, CMTM stages on stages 1 and 3. I think deeper analysis would be necessary. What patterns can be explained? Please provide deeper analysis, I think pairwise tests would be informative as well in addition to ANOVA.

2. For better readability, I suggest splitting the contents in the discussion section into several subsections, each titled with 1 specific finding.

3. On section 3.3, linear correlation was analyzed between CMTM expressions with the abundance of immune infiltrating cells. Several findings piqued my interest. Did you further measure the pairwise correlation between the CMTMs? For example, CMTM2,4,7,8 were found to possess significant positive correlation with Tregs. From my understanding, it is also possible that the expressions for some of the CMTMs are correlated as well, meaning that the association between Tregs with CMTM2 and 4 for example may occur simultaneously. For simplicity purposes, I think some colored scatterplots can be useful in analyzing this aspect.

**Reviewer #2:**  Thank you for a clear article, which outlines the rationale for the study and what was done, and how to interpret the findings.

I have a very minor suggestion:

page 6 - final sentence doesn't need to say p<0.05 twice.

Could 95% confidence intervals be added to the survival curves in the figures?

**Reviewer #3:**  1. The study investigates the CMTM family in esophageal carcinoma (ESCA), focusing on immune infiltration. However, similar analyses have been performed for other cancers. What novel insights specific to ESCA does this study provide compared to prior research?

2. The study uses TCGA and GTEx datasets, but there is no indication of validation on independent datasets or real-world clinical samples.

3. The manuscript focuses on the general ESCA population without stratifying between squamous cell carcinoma (ESCC) and adenocarcinoma (EAC), which are biologically distinct.

4. The manuscript highlights M2 macrophages and Tregs but does not provide sufficient mechanistic insights into how the CMTM family directly influences these immune cell populations.

5. The study relies heavily on bioinformatics analyses of TCGA and GTEx data without including experimental validations.

6. Correlations between CMTM expressions and immune infiltration are reported without controlling for potential confounders.

7. More references on bioinformatics-based workflow studies should be added to attract a broader readership i.e., PMID: 34572330, PMID: 37523012.

8. The markers used to identify M2 macrophages and Tregs (e.g., CD163, MS4A4A, VSIG4) are well-established, but the rationale for selecting these specific markers over others is not explained.

9. The manuscript uses P<0.05 as the significance threshold but does not address adjustments for multiple comparisons in large-scale data analyses.

10. While the manuscript suggests that CMTMs may be immunotherapy targets, it does not discuss how this can be translated into actionable therapies.

11. Without validation in clinical samples or patient-derived models, the translational relevance of the findings remains speculative.

12. The study associates higher M2 macrophage and Treg abundance with reduced survival. How do CMTMs compare with other prognostic markers in ESCA?

6. PLOS authors have the option to publish the peer review history of their article (what does this mean? ). If published, this will include your full peer review and any attached files.

**Do you want your identity to be public for this peer review?** For information about this choice, including consent withdrawal, please see our Privacy Policy .

Reviewer #1: **Yes: ** Gregorius Natanael Elwirehardja

Reviewer #2: No

Reviewer #3: No

---

## [Author Response · Author response to Decision Letter 1]

27 Jan 2025

Dear Reviewers

We would like to express our sincere gratitude to you for your valuable time and constructive comments on our manuscript. We have carefully considered all the suggestions and have made significant revisions to the manuscript accordingly. We provide a detailed response to each comments in “Response to Reviewers”.

We believe that the revisions have significantly improved the manuscript and hope that it now meets the high standards for publication in [PLOS ONE]. Thank you once again!

Liying Xue,

Laboratory of Pathology, Hebei Medical University

No. 361, Zhongshan Eastern Road, Shijiazhuang, China. 050000

xueliying123@163.com; xueliying@hebmu.edu.cn

---

## [Decision Letter · Decision Letter 1]

11 Feb 2025

PONE-D-24-50157R1Comprehensive analysis of CMTM family and immune infiltration in esophageal carcinomaPLOS ONE

Dear Dr. Xue,

Thank you for submitting your manuscript to PLOS ONE. After careful consideration, we feel that it has merit but does not fully meet PLOS ONE’s publication criteria as it currently stands. Therefore, we invite you to submit a revised version of the manuscript that addresses the points raised during the review process.

**Please  kindly address the concerns raised by Reviewer #2. Thank you.**

We look forward to receiving your revised manuscript.

Kind regards,

Dave Mangindaan

Academic Editor

PLOS ONE

**Journal Requirements:**

Reviewers' comments:

Reviewer's Responses to Questions

**Comments to the Author**

1. If the authors have adequately addressed your comments raised in a previous round of review and you feel that this manuscript is now acceptable for publication, you may indicate that here to bypass the “Comments to the Author” section, enter your conflict of interest statement in the “Confidential to Editor” section, and submit your "Accept" recommendation.

Reviewer #2: (No Response)

Reviewer #3: All comments have been addressed

2. Is the manuscript technically sound, and do the data support the conclusions?

Reviewer #2: Yes

Reviewer #3: Yes

3. Has the statistical analysis been performed appropriately and rigorously? 

Reviewer #2: Yes

Reviewer #3: Yes

4. Have the authors made all data underlying the findings in their manuscript fully available?

Reviewer #2: Yes

Reviewer #3: Yes

5. Is the manuscript presented in an intelligible fashion and written in standard English?

Reviewer #2: Yes

Reviewer #3: Yes

6. Review Comments to the Author

**Reviewer #2:**  Thanks for an interesting article.

I just wish to confirm that all the p-values quoted in the text and in the figures were the FDR p values? Often the one on the figures are not, particularly if p-values from different types of analyses are included. Currently in the text it seems like they are - so just confirming that. Sometimes in studies they quote an uncorrected and then a corrected (or adjusted) p-value. Would that be appropriate here (maybe only for most significant results?) to see the impact of the FDR adjustment?

**Reviewer #3:**  My previous comments have been addressed. Thus I'm happy to recommend this manuscript to be published in PLOS One.

7. PLOS authors have the option to publish the peer review history of their article (what does this mean? ). If published, this will include your full peer review and any attached files.

**Do you want your identity to be public for this peer review?** For information about this choice, including consent withdrawal, please see our Privacy Policy .

Reviewer #2: No

Reviewer #3: No

---

## [Author Response · Author response to Decision Letter 2]

17 Feb 2025

Dear Revierwers:

Thank you for your thoughtful comments and for highlighting the importance of clarifying the p-values used in our study. We appreciate your suggestion to ensure rigor in our statistical reporting. Thank you again for your valuable feedback, which has helped us improve the manuscript.

Liying Xue,

Laboratory of Pathology, Hebei Medical University

No. 361, Zhongshan Eastern Road, Shijiazhuang, China. 050000

xueliying123@163.com; xueliying@hebmu.edu.cn

---

## [Decision Letter · Decision Letter 2]

28 Feb 2025

Comprehensive analysis of CMTM family and immune infiltration in esophageal carcinoma

PONE-D-24-50157R2

Dear Dr. Xue,

We’re pleased to inform you that your manuscript has been judged scientifically suitable for publication and will be formally accepted for publication once it meets all outstanding technical requirements.

Kind regards,

Dave Mangindaan

Academic Editor

PLOS ONE

Additional Editor Comments (optional):

Reviewers' comments:

Reviewer's Responses to Questions

**Comments to the Author**

1. If the authors have adequately addressed your comments raised in a previous round of review and you feel that this manuscript is now acceptable for publication, you may indicate that here to bypass the “Comments to the Author” section, enter your conflict of interest statement in the “Confidential to Editor” section, and submit your "Accept" recommendation.

Reviewer #2: All comments have been addressed

Reviewer #3: All comments have been addressed

2. Is the manuscript technically sound, and do the data support the conclusions?

Reviewer #2: (No Response)

Reviewer #3: Yes

3. Has the statistical analysis been performed appropriately and rigorously? 

Reviewer #2: (No Response)

Reviewer #3: Yes

4. Have the authors made all data underlying the findings in their manuscript fully available?

Reviewer #2: (No Response)

Reviewer #3: No

5. Is the manuscript presented in an intelligible fashion and written in standard English?

Reviewer #2: (No Response)

Reviewer #3: Yes

6. Review Comments to the Author

Reviewer #2: (No Response)

Reviewer #3: My previous comments have been addressed. I think this study can meet the requirement for publication.

7. PLOS authors have the option to publish the peer review history of their article (what does this mean? ). If published, this will include your full peer review and any attached files.

**Do you want your identity to be public for this peer review?** For information about this choice, including consent withdrawal, please see our Privacy Policy .

Reviewer #2: No

Reviewer #3: No

---

## [Editor Report · Acceptance letter]

PONE-D-24-50157R2

PLOS ONE

Dear Dr. Xue,

I'm pleased to inform you that your manuscript has been deemed suitable for publication in PLOS ONE. Congratulations! Your manuscript is now being handed over to our production team.

Kind regards,

on behalf of

Assoc. Prof. Dave Mangindaan

Academic Editor

PLOS ONE